# The peer effects of resident stock market participation: Evidence from 2019 CHFS

**Zhijian Lin**[1], **Manyu Kong**[2]*, **Guoli Li**[3], **Xin Wang**[4]

**1** College of Economics and Management, Mianyang Teachers'College, Mianyang, 621000, China, **2** School of Finance, Southwestern University of Finance and Economics, Chengdu, 610074, China, **3** School of Marxism, Hechi University, Hechi, 547000, China, **4** School of Finance, Tianfu College of SWUFE, Mianyang, 621000, China

* manyu_kong@163.com

**Data Availability Statement:** The data used to support the findings of this study have not been made available according to the Law of the People's Republic of China on the Protection of Secrets and relevant copyright agreements of The

## Abstract

Limited resident's participation in the stock market has become a key constraint to the capital market development. Utilizing the 2019 China Household Financial Survey (CHFS) data, our paper designs probit models to examine the peer effects of residents' stock market participation and explore the intermediary mechanisms with a multiple intermediary model. We find that: (1) Resident involvement in stock market decision-making exhibits significant peer effects. (2) Heterogeneity analysis reveals that males and rural residents display more pronounced peer effects than females and urban residents. Additionally, middle-aged residents demonstrate more potent peer effects than their younger and older counterparts, with the intensity of peer effects correlating with education levels. (3) We observe that the peer effects of market participation operate by altering economic expectations and enhancing residents' financial literacy. (4) Further investigation establishes that individuals engaging in stock market investments manifest peer effects when deciding whether to diversify their stock portfolio. This study holds reference value for analyzing the impact of social interaction on financial behaviors and regulating individuals' financial conduct.

## 1. Introduction

Stock market participation is an essential research subject in household finance due to its significant implications. According to conventional investment theory, investors are recommended to hold a certain amount of stock investments. Participation in the stock market can diversify a household's asset portfolio, which has implications for future income, consumption smoothing, and risk management. However, research has discovered a "limited participation puzzle" in stock market participation(Georgarakos, 2011) [1]. This phenomenon is also observed in China. The proportion of stock assets in Chinese households is low, and the number of natural person investors in China's stock market was only 12.07% of China's total population of 1.4 billion in 2018 [data from the National Bureau of Statistics]. Additionally, the number was only 6.4% of households' financial assets comprise stocks, according to the "Survey on the Assets and Liabilities of China's Urban Residents' Households in 2019." This

People's Republic of China, the author may use the data for research with the approval of the Survey and Research Center for China Household Finance (CHFS), but shall not disclose the source data to any third party. The data to be accessible upon reasonable request via a non-author point of contact, China Household Finance Survey Center (CHFS) data service department related information: Office Address: No. 55, Guanghua Village Street, Qingyang District, Chengdu, Sichuan Province; E-mail address:contactus@chfs.cn; Office Tel: 028 87352095; Website: http://chfs. swufe.edu.cn/; Zip Code: 610074.

**Funding:** The author(s) received no specific funding for this work.

**Competing interests:** The authors have declared that no competing interests exist.

current situation is detrimental to the growth of China's stock market, as well as the optimization of residents' asset structure and their financial needs.

While stock market participation ostensibly results from individual decisions, expectations and benefits tied to such involvement can be shaped by the behavior of others within the same social group. This phenomenon is commonly referred to as the peer effects. It implies that individuals' decisions related to consumption, production, and investment activities are influenced by the actions of their peers, resulting in a tendency toward convergence in economic decision-making within the same group (Ulrich Horst et al., 2006; Ling et al., 2018) [2,3]. This trend is especially pronounced in the adoption of new practices and investment in risky projects (Bursztyn et al., 2014; Ng & Wu, 2010) [4,5]. Although current studies analyzing residents' participation in financial and stock markets primarily focuses on rational decision-making frameworks involving factors like income and costs, it is essential to acknowledge that, in practice, Chinese residents often require higher financial literacy and more accessible information channels, leading to limited rational decision-making abilities. Consequently, it becomes crucial to investigate residents' participation in the stock market from the perspective of the peer effects to analyze the characteristics of financial individual involvement and investment behavior.

This paper, based on the 2019 China Household Finance Survey (CHFS) data, utilises the probit model to empirically test the stock participation behaviour of Chinese residents through the lens of peer effects. Additionally, the study conducts heterogeneity analysis to determine which types of residents exhibit more significant peer effects in stock market participation. Furthermore, a mediation model is employed to investigate the underlying mechanisms of peer effects in residents' stock market participation. Finally, our findings show that peer effects are also manifested in residents' diversified stock investments. This study provides a reference point for assessing the impact of interactions within groups of residents on their financial habits and regulating financial behaviour in practice.

## 2. Literature review

There are numerous studies exploring the factors influencing household participation in financial markets from various perspectives. Family demographic variables, such as wealth and education levels, have an impact on family financial market participation, and families with higher assets and education tend to be more willing to participate in risky financial markets (Dohmen, 2010; Kaustia et al., 2023) [6,7]. Additionally, residents' subjective variables, including risk preference, social network, and private information, also influence household financial market participation (Guiso et al., 2008; Barberis et al., 2006; Li et al., 2017) [8–10]. Many scholars have found that there is a significant disparity between real asset allocation and theory, with many families not participating in the stock market. The phenomenon that the proportion of household investment in stocks is significantly lower than the theoretically optimal allocation to risky assets is known as the "limited participation" mystery of the stock market, and it is prevalent in developed countries like the United Kingdom and the United States and is also prominent in China (Guiso & Sodini, 2013; Markku et al., 2023) [11,12].

Research on the factors affecting stock market participation has mainly focused on three levels: individual heterogeneity, household heterogeneity, and environmental heterogeneity. Studies indicate that individual heterogeneity affects stock market participation, specifically by demographic characteristics like age, race, education level, and risk attitude. Family investment decisions are significantly influenced by these factors (Campbell, 2006; Xu et al., 2022) [13,14]. Education level, in particular, has a positive influence on stock market participation. (Van Rooij et al., 2012; Ting et al., 2022) [15,16], and an increase in financial knowledge also

promotes family participation in the stock market and increases stock asset allocation (Van Rooij et al., 2012; Travo et al., 2019) [17,18]. After controlling for levels of education and financial knowledge, heightened cognitive ability may lead to an increase in the proportion of stock asset allocation among families in urban areas. (Grinblatt et al., 2011; Sivaramakrishnan et al., 2017) [19,20]. Whilst it is true that some research has found a "bell-shaped" distribution in stock market participation rates across age groups (Guiso et al., 2000; Joelle et al., 2020) [8,21], it is not so obvious that stock market participation in China has a life-cycle characteristic. (Cui & Insook, 2019) [22]. In addition, it has been found that risk attitude has a significant impact on household financial asset composition, and an increase in risk preference will promote family stock market participation and increase stock holdings (Guiso et al., 2008; Böckerman et al., 2021) [8,23]. Furthermore, religious beliefs can also affect the proportion of residents holding stocks (Renneboog et al., 2012; Yulia & Chris, 2022) [24,25]. Regarding the impact of household heterogeneity on stock market participation, existing literature mainly focuses on the influence of income and wealth generation (Fischer et al., 2015) [26]. Stock market participation among residents increases with income (Jing & Xiaojun, 2019) [27] and demonstrates a significant "wealth effect". In addition, housing has a significant crowding-out effect on family stock market participation, reducing the probability and depth of family entry into the stock and risk capital market(Chen & Ji, 2017) [28]. Environmental heterogeneity also affects residents' stock market participation behavior, with related research focusing mainly on factors such as trading frictions, liquidity constraints, social interactions, information channels, and trust (Alan, 2005; Willen & Kubler, 2006; Li, 2014; Gao, 2019; Ahern et al., 2014) [29–33].

Non-market interactions, for example social interactions, are also an important aspect of individuals' financial decision-making. Qize et al's (2022) [34] study shows that social interactions influence residents' participation in the stock market, and 'relationships' influence households' participation in the stock market through information channels and social interactions. (Riha et al., 2022) [35]. Similarly, a greater degree of trust also encourages residents' involvement in the stock market, as confidence in listed firms meaningfully increases participation rates. (Yang et al., 2021) [36]. The "peer effects" is defined as the influence of economic behavior by individuals within the same community or specific geographical area. In other words, it means that individuals tend to adopt the behaviors of those they are close to (Ziyao et al.,2022) [37]. This mechanism not only includes active communication within social circles but also involves observation, imitation, and reference among unfamiliar individuals within a community (Kedia, 2015) [38]. The study of peer effects originated in the field of education and was primarily used to explain the influence of classmates on individual students' academic performance, non-cognitive abilities, and preferences (Patacchini et al., 2017) [39]. Currently, research on peer effects has been extended to fields such as economics, management, and finance, encompassing topics like company innovation, financial arrangements, and investment behavior (Park et al., 2017) [40], as well as individual participation in gambling, entrepreneurial behavior, and charitable donations (Park & Manchanda, 2015; Smith et al., 2015) [41,42]. Additionally, scholars have also focused on the impact of peer effects on individual deviant behavior, suggesting improvements through social multipliers to address unfavorable economic behaviors such as personal crime and corporate misconduct (Duflo & Saez, 2003; Mowen & Boman, 2018) [43,44].

Current research on stock market participation tends to operate within an analytical framework that prioritises market factors over individual rational decision making, overlooking the mutual influence of residents in stock market participation. While some scholars have examined the impact of trust, social networks, and social interaction on stock market participation, these studies primarily measure proactive communication and learning among intimates through interpersonal expenditures. They do not take into account behavioural consistency

based on the psychology of similarity seeking, passive information acceptance, or conscious imitation among unfamiliar intimates within a given space. Furthermore, these studies have yet to delve into the underlying mechanisms, resulting in insufficient practical implications.

## 3. Research hypotheses and theoretical demonstration

### 3.1. Research hypotheses

Social networks such as relatives, friends, neighbors, and colleagues serve as channels for mutual assistance when individuals face knowledge deficiencies, information asymmetry, and high risks. These networks can influence individuals' emotions, cognition, and decision-making (Felmlee & Faris, 2013; Chai et al., 2019) [45,46]. Based on existing research and theories, the following research hypothesis is proposed:

The reference group and social recognition theory suggests that people's stock market decisions are influenced by the comparable economic decisions of other members of their social group. On one hand, based on reference group theory, individuals tend to maintain consistency with their social group, as they share the same cultural background, values, and collective consciousness. Therefore, they consider the similar behavior of fellow group members as an important indicator for decision-making (Acevedo, 2009) [47]. On the other hand, according to social recognition theory, individuals acquire basic operational knowledge and common sense about stock market investing by observing, communicating and imitating the behaviour of other group members who participate in the stock market. Therefore, this enhances the practicality and convenience of participating in the stock market by providing a clear understanding of the potential advantages, disadvantages and risks of investing in the stock market, while at the same time improving an individual's financial literacy. Additionally, this approach mitigates the aversion that may arise from vague perceptions of stock market investment (Rima & David, 2015) [48]. Furthermore, as individuals often have limited information on stock market investment, they form their expectations by observing the stock market participation behavior of their surrounding residents. When more residents within a community participate in the stock market, it generates positive expectations for future stock market development, thus promoting residents' participation in the stock market. Based on the rational behavior theory, innovation diffusion theory, social cognitive theory, and existing research, the following research hypothesis is proposed:

**Hypothesis 1**: There is a peer effect in residents' stock market participation, meaning that residents' level of stock market participation is positively correlated with the stock market participation level of other members within their social group.

According to social recognition theory, individuals can acquire financial knowledge related to the stock market through observational and verbal learning from other stock market participants within their social group. They can gain an understanding of simple analytical techniques and relevant operational procedures for stock market investment, thereby enhancing their ability and willingness to participate in the stock market. Additionally, residents can enrich their information channels and improve their valuation capabilities for stock market investment by observing and following the stock market participation of neighboring residents within their community. This social learning process is particularly frequent when individuals have limited information and a lack of financial knowledge, resulting in a stronger spillover of financial knowledge within their social group. Based on this, the following hypothesis can be proposed:

**Hypothesis 2**: The financial proficiency of residents is improved by their social group members' stock market involvement through the spillover of financial knowledge, which consequently encourages their individual participation in the stock market.

According to the theory of emotional contagion, an individual's emotions can be influenced by the emotions of other members of their social group. Therefore, an individual's economic expectations can also be infected by the optimistic emotions of other members of their social group. Moreover, based on the strategy of conforming to the majority, individuals dynamically adjust their economic expectations to align with the expectations of the majority (English et al., 2012) [49]. Combining with the theory of stock market valuation, it can be inferred that when an individual's stock market expectations are influenced by the optimistic emotions of their social group, the level of optimism in their stock market expectations will significantly increase, thereby promoting their participation in the stock market. Therefore, the following hypothesis can be obtained:

**Hypothesis 3**: The involvement of fellow community members in the stock market can bolster an individual's optimism towards their own stock market expectations, ultimately incentivising them to participate in the market.

## 3.2. Theoretical demonstration

Based on the principle of utility maximization in two periods, this study logically proves the research hypothesis that there is a "peer effect" in residents' stock market participation. The logical proof of this theory is as follows: By constructing a two-stage utility maximization model, it can be deduced that the average level of stock market participation among other members in the same social group has a positive influence on an individual's probability of participating in the stock market. $\Phi$ ($\Phi = (\sum_{j \neq i} y_i)/(N-1) = n/(N-1)$) represents the average level of stock market participation of other residents in the group except individual $i$; N represents the total number of members of the group to which an individual belongs, where n represents the number of members of the group who have participated in the stock market; $stock_i$ represents whether individual $i$ participates in the stock market or not, and is a dummy variable where 1 indicates participation and 0 indicates non-participation. And $prob_{stock_{i=1}}$ represents the probability of an individual participating in the stock market. In this model, using the form of logistic function for reference, the probability function of individual participation in the stock market is expressed as follows:

$$prob_{stock_i=1} = \frac{1}{1 + e^{-(\alpha \Delta U)}} + \varepsilon \tag{1}$$

Where, $\Delta U$ represents the utility difference between individual participation in the stock market and non-participation in the stock market (Ahern et al., 2014; Mowen & Boman, 2018) [33,44]. $\varepsilon$ is the error term. We draws on the utility function of the constant relative risk coefficient, introduces transaction cost and financial knowledge spillover, and uses the change of total utility of two periods to describe $\Delta U$, in which the utility of individual participation in the stock market is expressed as follows:

$$u^1 = u_t^1 + u_{t+1}^1 + \varepsilon^1 \tag{2}$$

In the above formula, $u_t^1$ represents the utility of an individual's participation in the stock market during the $t$ period, while $u_{t+1}^1$ represents the utility of the second period (that is, $t+1$ period) after the individual participates in the stock market. $\varepsilon^1$ is the error term. The utility $u_t^1$ of period $t$ is mainly composed of three parts: the first part is the utility corresponding to the wealth change of individuals after participating in the stock market; the second part is the disutility brought by the transaction cost of individuals participating in the stock market; the third part is the disutility brought by the social deviation caused by the difference between the

average situation of individuals participating in the stock market and their groups participating in the stock market (Riha et al., 2022) [35]. In this model, the utility function of the constant relative risk coefficient is used to describe the utility function caused by wealth change, and the current utility function of individual participation in the stock market is expressed as follows by combining the transaction cost and social deviation of stock market participation:

$$u_t^1 = \frac{(W_T - W_I)^{1-\gamma}}{1-\gamma} + C + S + \varepsilon_t^1 \tag{3}$$

Where, $(W_T - W_I)^{1-\gamma}/(1-\gamma)$ represents the wealth utility, $W_T$ represents the total wealth of the individual in the $t$ period, $W_I$ represents the funds invested by the individual $i$ in the stock market in the $t$ period; $C(\Phi) = -c/(N\Phi)$ denotes the transaction cost of individual $i$ participating in the stock market (Yulia & Chris, 2022) [25]. This function indicates that the higher the average level of inter-group stock market participation, the stronger the spillover effect of financial knowledge, thus weakening the disutility caused by the transaction cost of stock market participation, where $c$ is the coefficient and greater than 0. Specifically, when the individual participates in the stock market for the first time, it needs to spend some time and energy to learn and understand the relevant information, principles and relevant operation processes of the stock market products, which will generate corresponding transaction costs and bring negative utility. $S(\Phi) = -\sigma(1-\Phi)^2$ denotes the social utility of individual $i$ participating in the stock market (Cui & Insook, 2019) [22]. This function represents the negative utility brought by the discrepancy between individual's stock market participation and the average participation level of the group. To avoid this situation, individual stock market participation decisions will try to align with the participation level of the group they belong to. Conversely, when the average participation level within the group is low, individual i will also choose to observe. That is, the disutility caused by social deviation after individual $i$ participating in the stock market, where the parameter >0. $\varepsilon_t^1$ is the error term.

When the utility of the $t+1$ period after the individual participates in the stock market is expressed as follows, the function $u_{t+1}^1$ of the $t+1$ period effect of the individual $i$ chooses to participate in the stock market is shown as follows:

$$u_{t+1}^1 = \frac{[(W_T - W_I) + (1 + E(R))W_I]^{1-\gamma}}{1-\gamma} + \varepsilon_{t+1}^1 \tag{4}$$

Where, $(1+E(R))W_I$ represents the expected return of individual $i$ in $t+1$ period after participating in the stock market. E(R) is the expected return rate of the stock portfolio estimated by the individual based on the observation of the stock market participation return of the surrounding residents in the t period (Joelle et al., 2020) [21]. $\varepsilon_{t+1}^1$ is the error term. This model assumes that individuals base their decisions on asset pricing models and estimate the $\hat{\beta}$ of holding a stock market portfolio based on observable investment returns from surrounding individuals. The accuracy of their estimated $\hat{\beta}$ increases as the decision-maker observes more samples (i.e., when more residents in the group participate). The true relationship between $\beta$ and $\hat{\beta}$ is represented by $\beta = \theta\hat{\beta}$, where θ is the coefficient of trustworthiness $\theta = e^{-\frac{\omega}{N\Phi}}$. Therefore, the individual's expected returns E(R) on the stock portfolio can be expressed as follows:

$$E(R) = r_f + \beta[E(R_m) - r_f], \; \beta = \theta\hat{\beta} \tag{5}$$

Based on Formulas (3) and (4) above, the prescription value of residents' participation in the stock market $u^1 = u_t^1 + u_{t+1}^1/(1+\rho)$ can be expressed as follows (where $\rho$ is the time

factor, and $\varepsilon^1$ is the error term equaling to the sum of $\varepsilon_t^1$ and $\varepsilon_{t+1}^1$):

$$u^1 = \frac{(W_T - W_I)^{1-\gamma}}{1-\gamma} - \frac{c}{(\mathrm{N}\Phi)} - \sigma\left(1 - \Phi\right)^2 + \frac{[(W_T - W_I) + (1 + E(R))W_I]^{1-\gamma}}{(1-\gamma)(1+\rho)} + \varepsilon^1 \quad (6)$$

Similarly, assuming that residents will convert the same wealth into deposits and other financial products when they do not participate in the stock market, the average return of $t+1$ period is the risk-free return of $t+1$ period. Therefore, the utility $u^0$ of the two periods of individuals who do not participate in the stock market or other financial products can be expressed as follows:

$$u^0 = \frac{(W_T - \mathrm{W}_I)^{1-\gamma}}{1-\gamma} + \frac{[W_T + W_I(1 + r_f)]^{1-\gamma}}{(1-\gamma)(1+\rho)} + \varepsilon^0 \quad (7)$$

By combining Formula (6) and Formula (7), the utility difference between individuals participating in the stock market and those not participating in the stock market $\Delta U$ can be deduced as follows, and $\varepsilon$ is the subtraction of $\varepsilon^1$ and $\varepsilon^0$:

$$\Delta U = -\frac{c}{(\mathrm{N}\Phi)}$$
$$- \sigma\left(1 - \Phi\right)^2 + \frac{[(W_T - W_I) + (1 + E(R))W_I]^{1-\gamma}}{(1-\gamma)(1+\rho)} - \frac{[W_T + W_I(1+r_f)]^{1-\gamma}}{(1-\gamma)(1+\rho)} + \varepsilon \quad (8)$$

Based on the above statements, the probability model of individual participation in the stock market can be expressed as follows:

$$prob_{stock_i=1} = \frac{1}{1 + e^{-(\alpha\Delta U)}}$$

$$\Delta U = -C(\Phi) - + \frac{[(W_T - W_I) + (1 + E(R))W_I]^{1-\gamma}}{(1-\gamma)(1+\rho)} - \frac{[W_T + W_I(1+r_f)]^{1-\gamma}}{(1-\gamma)(1+\rho)} + \varepsilon$$

$$C(\Phi) = \sigma(1 - \Phi)^2$$
$$C(\Phi) = \frac{c}{(\mathrm{N}\Phi)} \qquad\qquad\qquad \theta = e^{-\frac{\omega}{N\Phi}}$$
$$E(R) = r_f + \beta[E(R_m) - r_f]$$
$$\beta = \theta\,\hat{\beta}$$

In the above model, if it can be proved that the bias of individual stock market participation probability ($prob_{stock_i=1}$) to the participation average ($\Phi$) of other residents in the community to which the individual belongs is greater than 0, then it can be proved that there is a peer effect of individual stock market participation, thus logically proving the hypothesis of this paper.

Taking the first-order partial derivative of $prob_{stock_i=1}$ with respect to $\Phi$ yields:

$$\frac{\partial probprob_{stock_i=1}}{\partial\Phi} = \frac{\partial prob_{stock_i=1}}{\partial\theta} \times \frac{\partial\theta}{\partial\Phi} + \frac{\partial prob_{stock_i=1}}{\partial C} \times \frac{\partial C}{\partial\Phi} + \frac{\partial prob_{stock_i=1}}{\partial S} \times \frac{\partial S}{\partial\Phi} \quad (9)$$

Combined with the mathematical calculation of this model and existing research literature, it can be proved that the bias of $prob_{stock_i=1}$ for residents' income expectation ($\theta$), financial knowledge spillover (C) and social deviation (S) are all greater than 0 (Park et al., 2017; Riha

et al., 2022) [35,40]

$$\frac{\partial prob_{stock_i=1}}{\partial \theta} \times \frac{\partial \theta}{\partial \Phi} > 0; \ \frac{\partial prob_{stock_i=1}}{\partial C} \times \frac{\partial C}{\partial \Phi} > 0; \ \frac{\partial prob_{stock_i=1}}{\partial S} \times \frac{\partial S}{\partial \Phi} > 0 \quad (10)$$

Finally, $\partial prob_{stock_i=1}/\partial \Phi > 0$ can be obtained, thus proving that research hypothesis 1 is confirmed, as shown in Eq (9), that is, there is a peer effect of individual stock market participation. In addition, Formula (10) proves that the peer effect will affect individual stock market participation through three mechanisms: residents' income expectation ($\theta$), financial knowledge spillover (C) and social deviation (S), so hypothesis 2 and hypothesis 3 are confirmed.

## 4. Data and empirical specification

### 4.1. Data description

This study draws upon data from the 2019 China Household Finance Survey (CHFS) conducted by the China Household Finance Survey and Research Center. The China Household Finance Survey is a nationwide household survey project initiated by the Research Center at the Southwestern University of Finance and Economics. Employing a proportional-to-population size (PPS) sampling methodology, the survey selects samples from across the nation. It ensures that the sample contains sampling units with a variety of characteristics and that the sample structure is relatively similar to the population structure, thus effectively improving the precision of the estimates. In addition, stratified sampling allows for estimating each stratum's overall parameters and target quantities. Multiple measures have been implemented to control both sampling and non-sampling errors, ensuring the representation and accuracy of the sample data. The empirical data utilized in this paper encompasses 29 provinces and 345 counties nationwide, including 1054 urban and rural communities.

Table 1 shows the characteristics of samples. 50.90% of the questionnaire respondents are female. The respondents have an average age of 49.07, of which range from 46 to 60 years old, accounting 24.89%. Nearly 43.71% of the respondents have 9 years of schooling or below, 46.73% of the respondents have attended a high school (10–12 years), and less than 10% of people have 12 years of schooling and beyond. About 10.38% of interviewees stated that their household annual income is less than 30,000 Yuan, and 47.68% of them have an annual

**Table 1. Demographic profiles of the respondents.**

| Item | Category | Proportion |
|---|---|---|
| Gender | Male | 49.10% |
| | Female | 50.90% |
| Age(years) | <25 | 23.45% |
| | 25–35 | 12.36% |
| | 36–45 | 12.06% |
| | 46–60 | 24.89% |
| | >60 | 27.24% |
| Schooling years | 0-9years | 43.71% |
| | 10-12years | 46.73% |
| | >12years | 9.56% |
| Household annual income (thousand yuan) | <30 | 10.38% |
| | 30–60 | 10.26% |
| | 60–90 | 31.67% |
| | >90 | 47.68% |

income above 90,000 Yuan. This demographic characteristic respondents is basicly consitent with the data of China, indicating our data has representativity.

## 4.2. Identification of peer effects

Peer effects fall within non-market interactions (Manski, 2000) [50]. In empirical research, it is essential to differentiate between endogenous interaction, situational interaction, and contextual effects to precisely identify peer effects among similar non-market interactions. This study focuses on endogenous interaction, which typically explains an individual's behavior by taking into account the performance of other members within the same group, highlighting the influence and interaction among group members.

Entering the stocks market is an interesting decision that needs to be analysed. Kaustia and Knüpfer (2012) argue that investment success stories propagated in a peer effect explain to some extent the pattern of influx of new investors into the market. Entering the market requires careful analysis [51]. Peers' stock market participation behaviour can affect others' decisions to enter the market through various channels. Firstly, individuals may utilise peer outcomes to update their views on long-term fundamentals, like the equity premium. Nevertheless, the stock market is an exceptional area where learning about fundamentals from peer outcomes is challenging compared to more deterministic atmospheres. Investment results from peer stock market represent small data samples with multiple biases: the substantial random element in returns involves unobservable factors like investment skill and risk exposure (Gortner & Wheele, 2019) [52]. Secondly, it is not possible to directly observe peer outcomes and one must rely on indirect cues, such as verbal statements. If this selectivity exists and individuals are unsure if their peers participate in the market, communication not only discloses the outcome but also the status of their peers' participation(Feld & Zölitz, 2015) [53]. If individuals are worried about their personal wealth in comparison to their peers, this can encourage them to participate in the market (Svetlana et al., 2022) [54].

Situational interaction indicates the association between an individual's economic behavior and the external characteristics of their group, where similar external factors lead to similarities in residents' economic choices. Contextual effects demonstrate that residents exhibit similar economic decision-making when facing similar institutional environments. This paper addresses the issue of situational interaction by introducing shared external attributes like residents' income, education, age, etc., as control variables in the model. Furthermore, it controls and identifies contextual effects by incorporating the average conditions of other residents with similar traits within the same group and community characteristics (Brock & Durlaufd, 2002) [55]. Instrumental variables are also employed, and IV-probit estimation methods are applied to rectify measurement errors and reflexivity issues and omit variable bias in identifying the peer effect. Due to the distinctive household registration system in China, which to some extent constrains the unrestricted movement of the population, the estimation error stemming from self-selection in peer effects is relatively minor (Nie et al., 2015; Li et al., 2013) [56,57].

## 4.3. Model construction

To test the research hypothesis, a Probit model is constructed to examine whether there exist peer effects on residents' stock market participation. The model is specified as follows (Li et al., 2013; Manski, 2000) [50,57]:

$$Pr(stocki_i^C|stocki_i^C) = 1) = \Phi(\alpha_0 + \alpha_1 peerstock_{-i}^C + \alpha_2 X_i + \alpha_3 M_{-i} + \alpha_4 Z_i + prov) \qquad (11)$$

Among them, $stock_i{}^c$ is the dependent variable representing whether individual *i* in group C participates in the stock market. The explanatory variable is the same-group variable of individual's stock market participation, $peerstock_{-i}{}^c$. $X_i$ represents control variables related to individual characteristics, $M_{-I}$ represents a set of same-group variables related to individual characteristics, and $Z_i$ represents a set of variables related to the characteristics of the group to which the individual belongs.

## 4.4. Variables declaration

Dependent variable: Stock market participation. Stock market participation is defined as whether residents participate in the stock market. In the survey questionnaire, it corresponds to the question "Do you have a stock account?; 1.yes, 2. no". If the answer is "yes", it is defined as 1 (indicating that the sample resident has a stock account), or 0 (indicating that the sample resident doesn't have stock account).

Independent variable: The independent variable in this study is the average level of stock market participation among individuals outside the resident's group. It is important to note that in reality, residents typically belong to various types of groups, such as hometown groups, school groups, family, and various hobby clubs. To address the complexity of identifying the resident's group affiliation in empirical analysis, this study adopts the resident's community (including the village) as the research group to demonstrate the existence of the peer effects. The rationale behind using the community as the research group is that it serves as the primary place where residents engage in production, life, and social activities (Li et al, 2013; Yann et al, 2009; Nie et al, 2021) [57–59]. Additionally, it is the main channel through which residents receive administrative management and perceive changes in government policies (Im et al., 2016) [60]. Therefore, the community is considered the most important and primary group to which residents belong. The peer effects variable for stock market participation among sample residents is defined as the average stock market participation of neighboring residents within the same community, excluding the sample resident. It is calculated as follows: Firstly, this study divides residents living in the same community into one group. Then, when calculating the peer variable, it computes the average stock market participation of other residents in the same community, excluding the sample resident ($peerstock_i{}^C$).

$$peerstock_{-i}{}^C = \frac{(\sum_{j \neq i}^{N} stock_j{}^C - stock_i{}^C)}{(N-1)} \qquad (12)$$

Where, $stock_i{}^C$ indicates whether the surveyed residents participate in the stock market, $stock_j{}^C$ represents the stock market participation of other residents in the community except the sample residents, C represents the visited community, and N represents the total number of households participating in the visit in the visited community (Yan, 2018) [29].

Control variables: In order to accurately estimate the peer effects of residents' stock market participation, this study introduces three levels of control variables. The first level consists of individual characteristics variables, including gender, age, education, income, assets, liabilities, online account balance, and consumption expenditure of the residents themselves. The second level is introduced to address situational interaction in identifying the peer effects. It includes the average values of age, education, income, assets, liabilities, and online account balance of residents other than the sample resident. The third level involves introducing community variables to overcome contextual effects. These community variables include the health level of residents in the sample community, the proportion of young people in the community, and the average family size in the community.

The relevant variables used in this empirical study are listed in Table 2 below.

**Table 2. Description of variables.**

| | Variables | abbrev | items |
|---|---|---|---|
| Dependent variable | Stock market participation | $stock_{-i}^C$ | Sample stock market participation, corresponding to the question: "Do you have a stock account?" If the answer is "yes", it is defined as 1, otherwise is 0 |
| Independent variable | Stock market participation peer variables | $peerstock_{-i}^C$ | The average level of stock market participation of other residents in the same community except the sample, calculated by the averaging the stock market participation of other residents |
| Individual characteristic | Total income | $tincome_i^C$ | Total income, corresponding to the question: "How much is your total income?" (Unit: ten thousand yuan) |
| | Age | $age_i^C$ | The age of the respondent |
| | gender | $sex_i^C$ | The gender of the respondent |
| | education | $edu_i^C$ | The years of education of the respondent |
| | Total asset | $asset_i^C$ | The total assets of the respondent (Unit: ten thousand yuan) |
| | Total debt | $debt_i^C$ | The total debt of the respondent (Unit: ten thousand yuan) |
| | Value of third- payment | $evalue_i^C$ | The online platform balance of the respondent (Unit: ten thousand yuan) |
| | Total household expenditure | $pay_i^C$ | Total household expenditure, the question is "How much is your total household expenditure?" (Unit: ten thousand yuan) |
| Group characteristics variables | Group age | $peerage_{-i}^C$ | The average age of other residents in the same community except the sample |
| | Group education | $peeredu_{-i}^C$ | The average education years of other residents in the same community except the sample |
| | Group income | $peerincome_{-i}^C$ | The average income of other residents in the same community except the sample |
| | Group debt | $peerdebt_{-i}^C$ | The average debt of other residents in the same community except the sample |
| | Group internet value | $peerevalue_{-i}^C$ | The average balance of the Internet platform of other residents in the same community except the sample |
| | Group asset | $peertasset_{-i}^C$ | The average asset of the Internet platform of other residents in the same community except the sample |
| Community characteristic variable | Community health | $chealth^C$ | The physical health status of the residents in the communities visited |
| | Community family size | $cfscale^C$ | The average number of family members in the households in the communities visited |
| | Proportion of youth | $cyouthrate^C$ | The proportion of youth in the total population of the communities interviewed |

## 4.5. Descriptive statistics

The descriptive statistical results of the relevant variables are shown in Table 3 below.

# 5. Empirical results

## 5.1. Benchmark regression of peer effects of stock market participation

The baseline regression results for the peer effects of residents' stock market participation are reported in Table 4 (1)-(3). Table 4 (1) presents the regression results for individual characteristic variables of residents. The second column (2) includes the addition of individual characteristics variables in the same group, while the third column (3) incorporates community characteristic variables. As shown in Table 4 (1)-(3), the coefficient for stock market participation ($peerstock_i^C$) is significantly positive in each column, providing evidence for the existence of the peer effects in residents' stock market participation and confirming hypothesis 1, consistent with the findings of Bursztyn et al (2014) [4].

Analyzing the control variables, it can be observed that age, education level, income, assets, and household expenditure significantly promote residents' stock market participation. On the other hand, gender, liabilities, and third-party internet payments significantly decrease the probability of their involvement in stock market.

**Table 3. Descriptive statistics.**

| Variables | Mean | S.D | Min | Max | Obs |
|---|---|---|---|---|---|
| $stock_{-i}^{C}$ | 0.076 | 0.265 | 0 | 1 | 13027 |
| $peerstock_{-i}^{C}$ | 0.229 | 2.343 | 0 | 0.615 | 13027 |
| $gender_{i}^{C}$ | 0.491 | 0.500 | 0 | 1 | 13027 |
| $age_{i}^{C}$ | 49.070 | 14.410 | 20.000 | 75.000 | 13027 |
| $edu_{i}^{C}$ | 9.993 | 3.847 | 0.000 | 16.000 | 13027 |
| $tincome_{i}^{C}$ | 9.825 | 9.014 | 0.047 | 39.260 | 13027 |
| $asset_{i}^{C}$ | 122.700 | 160.400 | 4.485 | 696.000 | 13027 |
| $debt_{i}^{C}$ | 6.767 | 16.290 | 0 | 84.500 | 13027 |
| $evalue_{i}^{C}$ | 0.222 | 0.409 | 0 | 2.000 | 13027 |
| $pay_{i}^{C}$ | 10.270 | 8.930 | 1.456 | 43.190 | 13027 |
| $peerage_{-i}^{C}$ | 41.330 | 6.241 | 19.670 | 82.000 | 13027 |
| $peeredu_{-i}^{C}$ | 3.844 | 0.879 | 1.833 | 6.939 | 13027 |
| $peerincome_{-i}^{C}$ | 10.410 | 6.121 | 2.547 | 24.220 | 13027 |
| $peertasset_{-i}^{C}$ | 130.100 | 121.600 | 17.780 | 450.300 | 13027 |
| $peerdebt_{-i}^{C}$ | 9.591 | 16.690 | 0 | 268.000 | 13027 |
| $peerevalue_{-i}^{C}$ | 0.273 | 0.227 | 0.030 | 0.813 | 13027 |
| $chealth^{C}$ | 2.363 | 0.334 | 1.176 | 3.667 | 13027 |
| $cfscale^{C}$ | 4.050 | 0.984 | 2.000 | 8.286 | 13027 |
| $cyouthrate^{C}$ | 0.365 | 0.117 | 0 | 0.750 | 13027 |

## 5.2. Endogeneity and robustness tests

Due to potential measurement errors or omitted variables, the problem of endogeneity may be present in the results of the baseline regressions on residents' participation in the stock market. To address this endogeneity problem, this study employs an instrumental variable (IV) approach to test the endogeneity of peer effects on residents' stock market participation in the baseline regression (Nie et al., 2015; Li et al., 2013) [46,47]. In this part, the financial literacy of other neighboring residents in the same community ($peerfl_{-i}^{C}$) is used as the instrumental variable for the baseline regression, and the endogeneity is tested using the IV-probit model. On the one hand, financial literacy is positively correlated with the stock market participation of individual, affected by others' financial literacy in same community. So $peerfl_{-i}^{C}$ relates to other neighboring residents serves as the peer effects variable, which relevant to the independent variable ($peerstock_{-i}^{C}$) (Yin et al., 2014) [12]. On the other hand, the financial literacy of other neighboring residents, excluding the sample residents, does not affect the stock market participation of the sample, and uncorrelates with the random disturbance term. Hence, it is reasonable to using $peerfl_{-i}^{C}$ as the instrumental variable of $peerstock_{-i}^{C}$. We conduct an endogeneity test using the IV-probit model with the aforementioned instrumental variable, as shown in Table 5 (1). The coefficient of $peerstock_{-i}^{C}$ remains significant and positive, consistent with the coefficients and significance levels of the main variables. Additionally, the coefficient of the first-stage instrumental variable ($peerfl_{-i}^{C}$) is significant at a 5% level, and the first-stage F-statistic is greater than 15, thus ruling out the weak instrument problem, further confirming the accuracy of the baseline regression.

To validate the robustness of the baseline regression, this study employs a combination of methods including variable substitution, sensitivity analysis, and group regression. The specific testing process is as follows:

Firstly, the variable substitution method is used to test robustness. In the baseline regression, the average stock market participation of other residents in the same community,

**Table 4. Baseline regression results.**

| VARIABLES | (1) | (2) | (3) |
|---|---|---|---|
| | stock$_{-i}$$^C$ | stock$_{-i}$$^C$ | stock$_{-i}$$^C$ |
| peerstock$_{-i}$$^C$ | 0.1120*** | 0.1075*** | 0.1071*** |
| | (55.880) | (53.309) | (51.271) |
| gender$_i$$^C$ | -0.0068*** | -0.0045*** | -0.0045*** |
| | (-4.521) | (-2.992) | (-2.970) |
| age$_i$$^C$ | 0.0018*** | 0.0019*** | 0.0018*** |
| | (6.925) | (7.142) | (7.064) |
| edu$_i$$^C$ | 0.0090*** | 0.0070*** | 0.0070*** |
| | (32.256) | (24.586) | (24.547) |
| tincome$_i$$^C$ | 0.0005*** | 0.0003*** | 0.0003*** |
| | (5.642) | (3.287) | (3.396) |
| asset$_i$$^C$ | 0.0001*** | 0.0001*** | 0.0001*** |
| | (26.188) | (22.106) | (21.908) |
| debt$_i$$^C$ | -0.0001** | -0.0001** | -0.0001** |
| | (-2.059) | (-2.110) | (-2.087) |
| evalue$_i$$^C$ | -0.0050*** | -0.0050*** | -0.0051*** |
| | (-3.109) | (-3.150) | (-3.178) |
| pay$_i$$^C$ | 0.0009*** | 0.0009*** | 0.0009*** |
| | (11.240) | (11.050) | (11.066) |
| peerage$_{-i}$$^C$ | | -0.0001 | -0.0004 |
| | | (-0.420) | (-1.397) |
| peeredu$_{-i}$$^C$ | | 0.0233*** | 0.0233*** |
| | | (17.145) | (14.998) |
| peerincome$_{-i}$$^C$ | | -0.0000 | -0.0000 |
| | | (-0.220) | (-0.063) |
| peertasset$_{-i}$$^C$ | | -0.0000*** | -0.0000*** |
| | | (-3.735) | (-3.568) |
| peerdebt$_{-i}$$^C$ | | -0.0000 | -0.0000 |
| | | (-0.493) | (-0.655) |
| peerevalue$_{-i}$$^C$ | | 0.0152*** | 0.0156*** |
| | | (4.191) | (4.258) |
| chealth$^C$ | | | 0.0057* |
| | | | (1.731) |
| cfscale$^C$ | | | -0.0009 |
| | | | (-0.623) |
| cyouthrate$^C$ | | | -0.0125 |
| | | | (-0.849) |
| Pseudo R2 | 0.3324 | 0.3421 | 0.3422 |
| Obs | 13027 | 13027 | 13027 |

Note:

*, **, and *** indicate significance at the 10%, 5%, and 1% levels respectively, and the value in parenthesis is T-value, the same as follows.

excluding the sample residents, is used as the explanatory variable to examine the existence of peer effects on residents' stock market participation. In this step, we utilize the residents' social expenditure (*socialpay*) as a replacement variable for the aforementioned peer effects variable in the baseline regression. The results are shown in Table 5 (2). The coefficient of *socialpay*

**Table 5. Endogeneity and robustness tests.**

| VARIABLES | (1) | (2) | (3) | (4) | (5) |
|---|---|---|---|---|---|
| | $stock_i^C$ | $stock_i^C$ | $stock_i^C$ | $stock_i^C$ | $stock_i^C$ |
| $peerstock_{-i}^C$ | 0.9579*** | | | 1.2416*** | 1.2649*** |
| | (10.738) | | | (16.438) | (41.421) |
| socialpay | | 0.0167*** | | | |
| | | (3.387) | | | |
| $fpeerstock_{-i}^C$ | | | 0.0439 | | |
| | | | (1.113) | | |
| Control variables | Yes | Yes | Yes | Yes | Yes |
| Pseudo R2 | | 0.3294 | 0.3116 | 0.3572 | 0.3169 |
| Obs | 13027 | 13027 | 13027 | 5504 | 7523 |
| First-stage F value | 228.98 | | | | |
| $peerfl_{-i}^C$ | 0.4351*** | | | | |
| | (28.23) | | | | |
| Wald test | 22.50 | | | | |
| P | 0.0000 | | | | |

remains significant and positive, and the coefficients of the main variables are consistent with the baseline regression, thus confirming the robustness of the baseline regression.

Furthermore, to enhance the accuracy and robustness of the empirical analysis, this study employs a simulation test to further confirm the robustness of the baseline regression. This involves constructing a simulated community by randomly selecting residents from different communities within the same district as the sample. Then, using the same calculation method as before, a spurious peer effects variable ($fpeerstock_{-i}^C$) is computed for this simulated community. This spurious "peer effects variable" is then substituted for the original peer effects variable in the regression analysis. Finally, the significance of the coefficient of the simulated peer effects variable is observed. Table 5 reports the results of the simulation test in column (3), which show that the coefficients of the simulated peer effects variables are not significant, demonstrating the robustness of the baseline regression.

Finally, this study further validates the robustness of the baseline regression by conducting group regression analysis. Specifically, the total sample is divided into two groups based on whether the household size (i.e., the number of family members) is greater than the average household size. The group with a household size greater than the average is categorized as the large household size group, while the group with a household size smaller than or equal to the average is classified as the small household size group. The baseline regression model is then re-estimated for each group separately. The results are shown in Table 5 (4) and (5), where both the large household size group and the small household size group exhibit a significant positive association between the peer effects variable and stock market participation, consistent with the results of the baseline regression. This further confirms the robustness of the peer effects on stock market participation.

## 5.3. Heterogeneity analysis

In this section, the paper conducts a heterogeneity analysis from four aspects: gender, age, household registration characteristics, and education level, to study which residents exhibit stronger peer effects in stock market participation decisions.

**5.3.1. Heterogeneity analysis by genders and age.** To examine whether the peer effects in stock market participation differ across genders, this study divides the total sample into male

**Table 6. Heterogeneity analysis by gender and age.**

| VARIABLES | (1) | (2) | (3) | (4) | (5) |
|---|---|---|---|---|---|
| | Male | Female | Youth | Middle-aged | Elderly |
| | Probit | Probit | Probit | Probit | Probit |
| | $stock_i^C$ | $stock_i^C$ | $stock_i^C$ | $stock_i^C$ | $stock_i^C$ |
| $peerstock_{-i}^C$ | 0.1110*** | 0.0994*** | 0.0844*** | 0.1123*** | 0.0810*** |
| | (38.008) | (33.581) | (13.720) | (38.857) | (20.906) |
| Control variables | Yes | Yes | Yes | Yes | Yes |
| Pseudo R2 | 0.3696 | 0.3343 | 0.3174 | 0.3837 | 0.2384 |
| Obs | 6366 | 6661 | 2566 | 7155 | 3306 |

and female groups and conducts separate regressions using the baseline model. The results are presented in Table 6 (1) and (2), which show that significant peer effects exist in both the male and female groups. Additionally, the male group exhibits stronger peer effects in stock market participation compared to the female group. Moreover, to investigate whether the peer effects in stock market participation vary with age, this study categorizes the total sample into three age groups: youth group (aged between 18 and 35 years), middle-aged group (aged between 35 and 60 years), and elderly group (aged above 60 years). Subsequently, separate regressions are conducted for each age group using the baseline regression model. The results are presented in Table 6. As shown in columns (3)-(5) of Table 6, significant peer effects are observed in all age groups, and the strength of the peer effects first increases and then decreases as age increases. The findings of Kaustia et al (2023) [12] are in general agreement with the findings of this paper.

**5.3.2. Heterogeneity analysis of household registration and education level.** To investigate whether there are differences in peer effects on stock market participation between urban and rural residents, this study divides the total sample into urban residents and rural residents based on their household registration characteristics. Separate regressions are then conducted for each group using the baseline regression model. The results are presented in Table 7 (1) and (2), which show that significant peer effects exist in both the urban and rural groups. Furthermore, the rural group exhibits stronger peer effects compared to the urban group. Additionally, to examine whether the peer effects in stock market participation vary with education level, this study categorizes the total sample into five groups: primary school and below, middle school, high school or vocational school, junior college, and bachelor's degree and above. Subsequently, separate regressions are conducted for each education level group using the baseline regression model. The results are shown in Table 7 (3), (4), (5), (6), and (7). The regression

**Table 7. Heterogeneity analysis of household registration and education level.**

| VARIABLES | (1) | (2) | (3) | (4) | (5) | (6) | (7) |
|---|---|---|---|---|---|---|---|
| | Rural | Urban | Primary school and below | Middle school | High school or vocational school | Junior college | Bachelor's degree and above |
| | $stock_i^C$ | $stock_i^C$ | $stock_i^C$ | $stock_i^C$ | $stock_i^C$ | $stock_i^C$ | $stock_i^C$ |
| $peerstock_{-i}^C$ | 0.1692*** | 0.0244*** | 0.0210*** | 0.0525*** | 0.1120*** | 0.1148*** | 0.3126*** |
| | (41.831) | (12.562) | (12.040) | (18.290) | (29.805) | (10.362) | (30.361) |
| Control variables | Yes | Yes | Yes | Yes | Yes | Yes | Yes |
| Pseudo R2 | 0.3677 | 0.3379 | 0.3599 | 0.3482 | 0.3397 | 0.3173 | 0.2935 |
| Obs | 3404 | 9623 | 2163 | 3897 | 2980 | 1904 | 2083 |

results based on education level groups indicate that all groups exhibit peer effects in stock market participation. Among them, the group with a high school education level demonstrates the strongest peer effects, followed by the bachelor's degree and above group, junior college group, middle school group, and primary school and below group. The study of Xu et al (2022) [14] concluded that educational attainment is an important factor influencing stock market participation.

## 5.4. Analysis of impact mechanisms

In this part, we further explore the mechanism behind the peer effects on stock market participation among residents. According to social cognition theory, residents acquire basic knowledge and operational procedures to improve their financial literacy by observing, communicating, and imitating the financial behaviors of other residents. Financial literacy, in turn, can promote residents' stock market participation (Mellor, 2019) [61]. Additionally, based on the theory of expectations, individuals often adjust their economic expectations by observing the economic behavior of their surrounding peers due to imperfect and asymmetric information when participating in the economy (Ghiglino & Goyal, 2010) [62]. When neighboring residents' stock market participation increases within the same community, the group's optimistic expectations about the stock market also increase. Residents are usually influenced by the sentiment of the group and form positive expectations about the stock market, which further promotes stock market participation.

Therefore, a mediation model is designed to proved whether financial literacy ($fl_i^C$) and future market expectations ($expectmacro_i^C$) are the mediating variables of the peer effects on residents' stock market participation. Financial literacy ($fl_i^C$) is measured using factor analysis based on financial knowledge questions. The questions are: "Suppose the bank's annual interest rate is 4%, if you deposit \$100 in a bank for a fixed term of 1 year, the principal and interest you will get after 1 year will be?1. less than \$104, 2. equal to \$104, 3. greater than \$104, 4. impossible to calculate "; "Suppose the bank's interest rate is 5% per year and the inflation rate is 8% per year, what can you buy after a year of saving \$100 in the bank that will be? 1. more than a year ago, 2. as much as a year ago, 3. less than a year ago, 4. impossible to calculate"; "Which do you think is riskier in general, equity or debt funds? 1. equity funds, 2. debt funds, 3. never heard of equity funds, 4. never heard of debt funds, 5. neither, 6. the same";"Which do you think is riskier in general, Main Board stocks or GEM stocks?1. Main Board, 2. GEM, 3. never heard of Main Board stocks, 4. never heard of GEM stocks, 5. Neither heard of them, 6. the same ". If the answer to the above question is " impossible to calculate " or " never heard of either", the value is 0. Those questions cover the financial calculating ability, the knowledge about inflation and risk, and financial market knowledge of the sample. The validity and reliability of the resulting scale were then tested based on the responses to these questions. The Cronbach's alpha of these scales was 0.865, the KMO value was 0.786, and the Bartlett's test was significant; Besides, the component matrix loadings of each rotation were greater than 0.682, thus the validity and reliability of above scales measuring financial literacy. While, we quantify market expectations ($expectmacro_i^C$) with the expectation of residents to the employment market, real estate market and economic growth, which is obtained from questionnaire items.

The regression results are presented in Table 8. Specifically, Table 8 (2) shows that the average level of stock market participation among other residents in the same community increases residents' financial literacy. Liao et al (2017) [10] argue that individuals with higher financial literacy are more likely to hold risky financial assets. Table 8 (3) demonstrates that the average level of stock market participation among other residents in the same community increases

Table 8. Mechanism analysis of peer effects.

| VARIABLES | (1) | (2) | (3) | (4) |
|---|---|---|---|---|
| | $stock_i^C$ | $fl_i^C$ | $expectmacro_i^C$ | $stock_i^C$ |
| $peerstock_{-i}^C$ | 0.1071*** | 0.0026*** | 0.0227*** | 0.1823*** |
| | (51.271) | (4.167) | (6.962) | (28.484) |
| $fl_i^C$ | | | | 0.0393*** |
| | | | | (16.070) |
| $expectmacro_i^C$ | | | | 0.0210*** |
| | | | | (9.213) |
| Control variables | Yes | Yes | Yes | Yes |
| Pseudo R2/R2 | 0.3422 | 0.0612 | 0.1538 | 0.3352 |
| Obs | 13027 | 13027 | 4028 | 4028 |

residents' optimistic expectations about the stock market, in general agreement with the findings of Sivaramakrishnan et al (2017) [20]. Table 8 (4) confirms that residents' financial literacy promotes their stock market participation, and optimistic expectations about the stock market also enhance stock market participation. Taken together, the empirical results in Table 8 (1)-(4) provide evidence that the average level of stock market participation among other residents in the same community promotes residents' stock market participation by enhancing their financial literacy and improving their economic expectations. Therefore, hypotheses 2 and 3 are proved.

## 6. Further study

As mentioned earlier, this study has established the presence of peer effects in residents' stock market participation and identified two mechanisms: financial literacy improvement and expectation adjustment. In this section, the study aims to examine whether there are also peer effects on diversification decisions when residents invest in the stock market. To investigate this, a dummy variable, $investv_i^C$ (diversification status of residents' investments), is introduced to measure whether residents engage in diversification when participating in the stock market. If residents hold a portfolio of stocks with two or more different types of stocks, it indicates that they practice diversification when investing in the stock market, and $investv_i^C$ is defined as 1. Otherwise, $investv_i^C$ is defined as 0 (Goetzmann & Kumar, 2008) [63]. Then, based on the previous calculation method for the peer effects variable, we calculate the corresponding peer effects variable $peerinvestv_{-i}^C$ for diversification decisions. $peerinvestv_{-i}^C$ is defined as follows:

$$peerinvestv = \frac{\left(\sum_{j \neq i}^N investv_j^c - investv_i^c\right)}{(N-1)} \quad (13)$$

In this case, $investv_i^C$ indicates whether the resident diversifies when participating in the stock market, $investv_j^C$ represents the diversification decisions of other residents in the same community excluding the sampled resident, variable C represents the community being surveyed, and variable N represents the total number of households participating in the survey in that community. We then conduct empirical analysis using the following model:

$$Pr(investv_i^C | investv_i^C) = 1) = \alpha_0 + \alpha_1 peerinvestv_{-i}^C + \alpha_2 X_i + \alpha_3 M_{-i} + \alpha_4 Z_i + province \quad (14)$$

Where, $investv_i^C$ is the dependent variable and represents whether resident i in community C engages in diversified investment when participating in stock market. The main explanatory

**Table 9. Further research empirical results.**

| VARIABLES | (1) | (2) | (3) | (4) |
|---|---|---|---|---|
| | $investv_i^C$ | $investv_i^C$ | $investv_i^C$ | $investv_i^C$ |
| $peerinvestv_{-i}^C$ | 0.1016** | 1.7521*** | 0.2609*** | 0.0812*** |
| | (2.558) | (3.555) | (2.630) | (1.966) |
| Control variables | Yes | Yes | Yes | Yes |
| Pseudo R2 | 0.2102 | | 0.1962 | 0.2339 |
| Obs | 1148 | 990 | 299 | 849 |
| First-stage F value | | 78.68 | | |
| $peereinvest_{-i}^C$ | | -3.089*** | | |
| | | (-2.55) | | |
| Wald test | | 3.25 | | |
| P | | 0.0713 | | |

variable is the peer effects variable for diversified investment, $peersinvestv_{-i}^c$. $X_i$ represents control variables related to individual characteristics, $M_{-i}$ represents a group of peer effects variables related to sampled residents' characteristics, and $Z_i$ represents a group of variables related to the characteristics of the village where the residents are located, such as in the baseline regression.

The empirical results in this case are presented in Table 9. Columns (1) represent the baseline regression model mentioned above. Column (1) includes fixed effects at the regional level. In the columns, the peer effects variables are significantly positive, indicating that residents' diversification behavior in stock market investments exhibits peer effects. Column (2) introduces the participation of other residents in the same community in internet financial investment as an instrument variable to test for endogeneity using the IV-probit model. Internet financial investment participation can crowd out the amount of investment allocated by residents to the stock market, reducing the number of stocks held by residents, and thereby meeting the "relevance" requirement of instrumental variables. Simultaneously, the participation of neighboring residents in internet financial investment does not crowd out the stock market investment of the sampled residents, satisfying the "homogeneity" requirement of instrumental variables. Therefore, it is reasonable to use the participation of other residents in internet financial investment ($peereinvest_{-i}^C$) as an instrument variable. Additionally, columns (3) and (4) present regression results based on whether the household size of the sampled residents is greater than the average household size. The corresponding coefficients of the main variables in these two groups align with the direction and significance level of those in (1), confirming the robustness of the empirical results.

## 7. Conclusions and implications

Using data from the 2019 China Household Finance Survey and Research Center, our paper examines the peer effect on residents' stock market participation in China and finds the following conclusions: First, a significant peer effect is observed in individuals' stock market participation. This means that the average level of stock market participation of other residents in the same community or village positively influences the individual's decision to participate in the stock market. Second, the strength of peer effects in residents' stock market participation varies across demographic groups. Male and rural residents show stronger peer effects than their female and urban counterparts. In addition, middle-aged residents show stronger peer effects than their younger and older counterparts. Among residents with different levels of education, those with a high school education show the strongest peer effects. Third, the stock market

participation of other residents within the same community contributes to residents' stock market participation by mediating improvements in financial literacy and raising stock market expectations. This suggests that residents' financial literacy and optimistic stock market expectations can be positively influenced by observing and imitating their neighbours' stock market participation. Finally, there are peer effects on residents' decisions to diversify their investments in the stock market.

Based on the conclusions, the study holds implications for financial practices. Firstly, it highlights peer effects on residents' financial behaviors, particularly in financial management. This insight can effectively address issues such as insufficient participation in commercial insurance and internet finance and financial self-exclusion among Chinese residents, especially in rural areas. Leveraging the interactive peer effects mechanism can significantly enhance rural households' economic participation. Secondly, financial institutions and regulatory authorities should be attentive to the interactions among residents' financial behaviors. On one hand, they can foster positive financial behaviors by harnessing peer effects. On the other hand, leveraging these peer effects can help mitigate negative financial behaviors, thereby aiding in regulating residents' financial conduct and reducing systemic risks in the financial system. Lastly, when formulating economic policies, government financial departments should consider the peer effects within communities, as it can enhance policy effectiveness through social multiplier effects.

## Author Contributions

**Data curation:** Zhijian Lin, Guoli Li, Xin Wang.

**Investigation:** Zhijian Lin, Manyu Kong.

**Writing – original draft:** Manyu Kong.

**Writing – review & editing:** Zhijian Lin, Manyu Kong.

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
