## [Decision Letter · Decision Letter 0]

19 Oct 2023

PONE-D-23-25473The peer effects of individual stock market participation: Evidence from 2019 CHFS Household Finance SurveyPLOS ONE

Dear Dr. Kong,

Thank you for submitting your manuscript to PLOS ONE. After careful consideration, we feel that it has merit but does not fully meet PLOS ONE’s publication criteria as it currently stands. Therefore, we invite you to submit a revised version of the manuscript that addresses the points raised during the review process.

.

We look forward to receiving your revised manuscript.

Kind regards,

Wajid Khan

Academic Editor

PLOS ONE

Journal Requirements:

2. You indicated that ethical approval was not necessary for your study. We understand that the framework for ethical oversight requirements for studies of this type may differ depending on the setting and we would appreciate some further clarification regarding your research. Could you please provide further details on why your study is exempt from the need for approval and confirmation from your institutional review board or research ethics committee (e.g., in the form of a letter or email correspondence) that ethics review was not necessary for this study? Please include a copy of the correspondence as an ""Other"" file.

Reviewers' comments:

Reviewer's Responses to Questions

**Comments to the Author**

1. Is the manuscript technically sound, and do the data support the conclusions?

Reviewer #1: Partly

Reviewer #2: Yes

2. Has the statistical analysis been performed appropriately and rigorously? 

Reviewer #1: N/A

Reviewer #2: Yes

3. Have the authors made all data underlying the findings in their manuscript fully available?

Reviewer #1: No

Reviewer #2: Yes

4. Is the manuscript presented in an intelligible fashion and written in standard English?

Reviewer #1: No

Reviewer #2: Yes

5. Review Comments to the Author

Reviewer #1: 1. The utility functions, such as equation 2, should include an error term.

2. Further elaboration, clarification, or supporting literature is needed to explain the specification or structure of the utility function, such as equation 2. Why do people only care about utility in two periods? People are supposed to care about their lifecycle utility.

3. There are some Chinese characters below equation 8.

4. More elaboration, clarification, or supporting literature is needed to justify the operationalization of the concept of community or peer.

5. Who is responsible for financial decision-making within households: the husband, wife, or is it a joint decision?

6. The choice set that includes only two options is unrealistic. For example, some households may have stock accounts in name only, but they do not use these accounts.

7. Opening an account ten years ago is not the same as opening an account today. The relevant community/peer is also dynamic.

8. The English language needs significant improvement.

Reviewer #2: Abstract should include an over view of research methodology adopted in research paper.

There is a grammatical mistake in 10th line of abstract i.e. “thed stock”.

Elaborate the introduction section by providing a comprehensive discussion about individual stock market participation. Provide a strong case in this section as to why individual stock market participation is needed. Also provide problem statement and need of carrying out this research study with strong recent justifications from literature. Author old reference citations from literature, the problem statement should be further elaborated by providing most recent citations from literature review.

Provide literature support for hypothesis development for each individual stock market participation factor of the research study. Provide recent and updated reference from literature review.

Provide justification that why only “China Household Finance Survey and Research Center (CHFS) in 2019” has been used as a database. Why some other most recent survey are not taken as a database.

Provide justification of why “three-stage, stratified, proportional-to-population size (PPS) sampling method” is used for data collection.

Operational definitions of variables used in the research study should be provided

There is no discussion of diagnostic statistics used in data analysis section. Author must incorporate various diagnostic statistics methods in order to understand various characteristics of data used for analysis.

Discussion in the data analysis section should be supported with references from literature review and acceptance and rejection of hypothesis should be properly highlighted.

There should be uniformity in the reference style given in references section of the research study.

6. PLOS authors have the option to publish the peer review history of their article (what does this mean?). If published, this will include your full peer review and any attached files.

Reviewer #1: No

Reviewer #2: No

---

## [Author Response · Author response to Decision Letter 0]

6 Jan 2024

The response to the editor and reviewers' Comments on PONE-D-23-25473

We thank both the editor and the reviewers for the careful and thorough reviews of our paper “The peer effects of individual stock market participation: Evidence from 2019 CHFS ” (manuscript ID: PONE-D-23-25473). 

Based on the comments, we made the following major changes. 

(1) To make the logic of the literature review and the research hypothesis clearer, we have redesigned the structure of these parts.

(2) We further describe the questionnaire allocation method, regional coverage and risk preference classification of sample data to show that the sample data in the paper is widely representative.

(3) To illustrate the more detailed verification process of mediation effect, we have revised the part of mechanism analysis.

(4) To make the main work, the contributions and the implication of this paper more detailed and closely related to the empirical results, we have revised these parts.

(5) For other issues in the article, such as misuse of conjunctions, spelling or citation errors, and lack of references, we made detailed responses and revisions to each of them. 

We thank you for your informative and constructive comments. We found them to be of great help. Per your suggestions, we have carefully revised our manuscript, and the revised part is marked in red in our manuscript and response. Besides, in what follows in the response, your comments are shown in blue, which are then followed by our point-by-point responses. 

The Responses to Reviewer 1's Comments

1. The utility functions, such as equation 2, should include an error term.

Response:

Thanks very much for your constructive comments. We've added the error term, marked with red color, to the utility function of the article, marked with red colour.

2. Further elaboration, clarification, or supporting literature is needed to explain the specification or structure of the utility function, such as equation 2. Why do people only care about utility in two periods? People are supposed to care about their lifecycle utility.

Response:

Thank you very much. Based on your comment, we have added corresponding literatures to elaborate the utility function.After careful consideration, we believe that the two-period model is more reasonable for analyzing residents' stock participation behavior:

1.In reality, most residents are irrational; their market information and financial capability are lacking, and there is apparent aversion to ambiguity before stock market participation, so they cannot consider the income trade-offs in many future periods brought by the current stock market participation decision as theoretically thought（Rooij et al.,2011;Sharma &Sood,2022).

2.Stock market participation is complex, but there is a difference between its participation and investment behavior. In the case of stock investing, such as picking stocks, adding stocks, or selling stocks, these make more sense considering the entire life cycle. But in contrast, the decision of participation behavior is more about comparing the utility gap between participation and non-participation, so the two-period is more in line with reality (Bekhet &Matar,2012;Gábor-Tóth et al.,2018).

3.Another reason for choosing a two-period model in this paper is to try to eliminate the interference of other factors, simplifying the interference of unnecessary factors and facilitating the argumentation.

Reference

[1] Rooij M V , Lusardi A , Alessie R .Financial literacy and stock market participation[J].Journal of Financial Economics, 2011, 101(2):449-472.DOI:10.1016/j.jfineco.2011.03.006.

[2]Sharma V K , Sood D .Behavioural Biases in Investor's Stock Market Participation in Post Pandemic Phase: A Literature Review Approach[J].ECS transactions, 2022(1):107.

[3]Bekhet H A , Matar A .Risk-Adjusted Performance: A two-model Approach Application in Amman Stock Exchange[J].International Journal of Business & Social Science, 2012, 3(7):34-45.DOI:doi:http://dx.doi.org/.

[4] Gábor-Tóth, Enikő, Georgarakos D .Economic policy uncertainty and stock market participation[J].CFS Working Paper Series, 2018.DOI:10.2139/ssrn.3006651.

3. There are some Chinese characters below equation 8.

Response:

Thank you very much, and we are so sorry about this written mistakes. We have correct this mistake, and check our paper carefully to avoid those mistakes. 

4. More elaboration, clarification, or supporting literature is needed to justify the operationalization of the concept of community or peer.

Response:

Thank you for your constructive suggestion. We have supplemented the literature to support the reasonability to choose the respondents’ community as the peer group. The literatures are listed as follows.

[1]Yann,Bramoullé,and,et al.Identification of peer effects through social networks[J].Journal of Econometrics, 2009.DOI:10.1016/j.jeconom.2008.12.021.

[2]Li Q , Zang W , An L .Peer effects and school dropout in rural China[J].China Economic Review, 2013, 27(Complete):238-248.DOI:10.1016/j.chieco.2013.04.002.

[3]Nie P , Wang L , Sousa-Poza A .Peer Effects and Fertility Preferences in China: Evidence from the China Labor-Force Dynamics Survey[J].The Singapore Economic Review, 2021.DOI:10.1142/S0217590821500120.

[4]Im H J , Kang Y ,Young Joon Park∗.Policy Uncertainty and Peer Effects: Evidence from Corporate Investment in China[J].Ssrn Electronic Journal, 2016.DOI:10.2139/ssrn.2713102.

5. Who is responsible for financial decision-making within households: the husband, wife, or is it a joint decision?

Response:

Thank you for your serious-minded review. The head of the household is defined as who plays a decisive role in family affairs.The survey questionnaire asked the following questions about the head of household: Which of the following is the head of your household? The head of household here refers to the person who plays a decisive role in family affairs, not necessarily the head of household in the household register. Thus, the gender of the head of the household is uncertain and refers specifically to a member of the family.

6. The choice set that includes only two options is unrealistic. For example, some households may have stock accounts in name only, but they do not use these accounts.

Response:

Thank you for your serious-minded review and constructive suggestion. As you say, there are flaws in using whether or not residents have stock accounts to fully measure the stock investment behavior of farm households. 

Ultimately, however, our rationale for choosing this indicator is as follows:On the one hand, owning an account is the first step for residents to participate in the stock market, and it is also a behavior that can be observed objectively, which can describe whether residents participate in the stock market more objectively than other stock market behaviors. Based on your suggestion, we compared the empirical data in this paper and found that 98% of the residents who own stock accounts have bought and sold stocks. On the other hand, the purpose of this paper is to analyze whether there is a cohort effect in residents' participation in the stock market, and the cohort effect is usually mediated by three intermediaries: expectation, information, and knowledge, which play an essential role in residents' account opening decisions. Therefore, it is reasonable to use whether or not to have an account as a variable to test the cohort effect of stock market participation. In addition, limited by data availability, other better indicators do not yield sufficient samples.

7. Opening an account ten years ago is not the same as opening an account today. The relevant community/peer is also dynamic.

Response:

Thank you for your professional comment. The actor's community is usually chosen as the interaction group in existing studies of peer effects. On the one hand, residents are in various groups, most of which are dynamic and changing. Due to China's unique household registration system and the cultural characteristic of settling down, community members in their cities and villages are more stable. On the other hand, the empirical analysis step of this paper is to test whether individual stock market participation and other residents in the same community are correlated through a baseline regression and then demonstrate its robustness through the instrumental variables method, replacement variables, and dummy tests, as well as group regression.

However, as the reviewer notes, this would ignore community residents' dynamic and changing characteristics. The unavailability of data stops us from realizing the dynamic panel analysis to cope with it. It has to be admitted that this is indeed one of the shortcomings of this paper, and we will keep exploring better ways to overcome it in future research.

8. The English language needs significant improvement.

Response:

Thank you so much for your careful reading and comment. We have polished our language according your advice.

The Responses to Reviewer 2's Comments

1.Abstract should include an over view of research methodology adopted in research paper.

Response:

Thanks very much for your constructive comments. According to your comment, the econometric methods used are explained in the abstract as follows, marked with red colour.

“Limited resident's participation in the stock market has become a key constraint to the capital market development. Utilizing the 2019 China Household Financial Survey (CHFS) data, our paper designs probit models to examine the peer effects of residents' stock market participation and explore the intermediary mechanisms with a multiple intermediary model.” 

2.There is a grammatical mistake in 10th line of abstract i.e. “thed stock”.

Response:

We are so sorry about this written mistakes. We have correct this mistake, and check our paper carefully to avoid those mistakes. 

3.Elaborate the introduction section by providing a comprehensive discussion about individual stock market participation. Provide a strong case in this section as to why individual stock market participation is needed. Also provide problem statement and need of carrying out this research study with strong recent justifications from literature. Author old reference citations from literature, the problem statement should be further elaborated by providing most recent citations from literature review. 

Response:

Thank you for your precious advice.

Investment in the stock market is of great practical significance. Individuals holding stocks can diversify their household asset portfolios, affecting their future earnings, consumption smoothing, and risk management, but also relating to the economic functioning of society as a whole and the stable development of financial markets. Traditional investment theory suggests that investors should hold a certain percentage of equity investments (Guiso, Haliassos, Jappelli, & Claessens, 2003). From the perspective of household asset allocation, a higher proportion of risk-free financial assets leads to lower financial returns than inflation for residents. Especially during periods of monetary policy easing, households are less resilient to inflation risk. Therefore, to further improve the allocation of the family's "pocketbook" and share the dividends of economic development through participation in the capital market not only responds to the fast-growing demand for residents' financial management in recent years but also enhances residents' consumption ability and drives the "long-term plan" of consumption upgrading (Huirong, 2020; Huirong, 2020; Huirong, 2020). Huirong, 2020; Kadoya et al, 2017). It is of great significance to explore the phenomenon of "limited participation" in the stock market of Chinese residents and to solve the problem of low participation of families in the stock market to stabilize and increase the property income of the residents and to let the residents enjoy the fruits of the development of the capital market.

To explain the phenomenon of "limited participation" in the stock market, theoretical and empirical research has been considered to be mainly affected by four factors—first, the family's wealth and income level. The second is a risk attitude. The third is macro-demographic, socio-economic, cultural, and social factors—the fourth is micro factors such as marital status, level of education, and financial knowledge. Although studies have explained the phenomenon of "limited participation" in the stock market from various perspectives, the relationship between peer efficacy and family stock market participation in China has not been directly discussed (Kaustia et al., 2023).

The paper cites the latest literature on the cohort effect of household stock market participation in China in the literature review. Therefore, we modified the introduction section and marked it in red.

Stock market participation is an essential research subject in household finance due to its significant implications. According to conventional investment theory, investors are recommended to hold a certain amount of stock investments. Participation in the stock market can diversify a household's asset portfolio, which has implications for future income, consumption smoothing, and risk management. However, research has discovered a "limited participation puzzle" in stock market participation(Georgarakos, 2011) [1]. This phenomenon is also observed in China. The proportion of stock assets in Chinese households is low, and the number of natural person investors in China's stock market was only 12.07% of China's total population of 1.4 billion in 2018 [data from the National Bureau of Statistics]. Additionally, the number was only 6.4% of households' financial assets comprise stocks, according to the "Survey on the Assets and Liabilities of China's Urban Residents' Households in 2019." This current situation is detrimental to the growth of China's stock market, as well as the optimization of residents' asset structure and their financial needs.

[1] Liu H .Housing Investment, Stock Market Participation and Household Portfolio choice: Evidence from China's Urban Areas[J].Papers, 2020.DOI:10.48550/arXiv.2001.01641.

[2] Kadoya Y , Khan M , Rabbani N .Does Financial Literacy Affect Stock Market Participation?[J].SSRN Electronic Journal, 2017.DOI:10.2139/ssrn.3056562.

[3] Kaustia M ,Knüpfer, Samuli.Peer Performance and Stock Market Entry[J].Social Science Electronic Publishing[2023-12-10].DOI:10.2139/ssrn.1359006.

4.Provide literature support for hypothesis development for each individual stock market participation factor of the research study. Provide recent and updated reference from literature review.

Response:

Thank you for your precious advice.To provide literature support for the hypothesis of each stock market participation factor in the study. Provide the most recent references in the literature review.We added the latest literature to the research hypothesis, listed as follows.

[1] Middleton L , Hall H , Raeside R .Applications and applicability of Social Cognitive Theory in information science research[J].SAGE PublicationsSage UK: London, England, 2019(4).DOI:10.1177/0961000618769985.

[2]Tang, W. Conducting in-Depth Research on Influencing Mechanisms for Improving Social Media Services Using Cognition Theory[J].Journal of Social Service Research,2023.,49, 511 - 529.

[3] Dotolo D , Lindhorst T , Kemp S P ,et al.Expanding Conceptualizations of Social Justice across All Levels of Social Work Practice: Recognition Theory and Its Contributions[J].Social Service Review, 2018, 92(2):143-170.DOI:10.1086/698111.

[4] Hammad M A .The Theory of Mind and Cognitive and Affective Empathy as Predictors of Proactive and Reactive Aggression of Hearing-Impaired and Normal Children[J].Journal of Educational and Psychological Sciences, 2017, 18:625-672.DOI:10.12785/JEPS/180420.

5.Provide justification that why only “China Household Finance Survey and Research Center (CHFS) in 2019” has been used as a database. Why some other most recent survey are not taken as a database.

Response

Thank you so much. Lack of data availability has been a limitation of household finance research. Compared to other data, China Household Finance Survey and Research Center (CHFS) in 2019 is the latest and best data available to us, whose sample size is sufficient enough and has good accuracy and representativenes.

6.Provide justification of why “three-stage, stratified, proportional -to- population size (PPS) sampling method” is used for data collection. 

Response

Thank you so much for your valuable comment. The three-stage, stratified, proportional-to-population size (PPS) sampling method is operated as follows. First, 2,585 counties throughout China were selected to be surveyed, excluding Tibet, Xinjiang, Inner Mongolia, Hong Kong, and Macao. Then, choose the aimed village from those counties. The final selection of households to be interviewed in a community is weighted by the number of people (or households) in that sampling unit. It ensures that the sample contains sampling units with a variety of characteristics and that the sample structure is relatively similar to the population structure, thus effectively improving the precision of the estimates. In addition, stratified sampling allows for estimating each stratum's overall parameters and target quantities. We have added the content of advantages of The three-stage, stratified, proportional-to-population size (PPS) sampling method in “4.1. Data description”, marked with red colour as follows.

It ensures that the sample contains sampling units with a variety of characteristics and that the sample structure is relatively similar to the population structure, thus effectively improving the precision of the estimates. In addition, stratified sampling allows for estimating each stratum's overall parameters and target quantities.

7.Operational definitions of variables used in the research study should be provided 

Response:

Thanks very much. We have describe the operational definitions of variables in detail in “5.4. Analysis of impact mechanisms” as follows, marked with red colour.

Therefore, a mediation model is designed to proved whether financial literacy (fliC) and future market expectations (expectmacroiC) are the mediating variables of the peer effects on residents' stock market participation. Financial literacy (fliC) is measured using factor analysis based on financial knowledge questions[ The questions are: "Suppose the bank's annual interest rate is 4%, if you deposit $100 in a bank for a 

fixed term of 1 year, the principal and interest you will get after 1 year will be?1. less than $104, 2. equal 

to $104, 3. greater than $104, 4. impossible to calculate "; "Suppose the bank's interest rate is 5% per 

year and the inflation rate is 8% per year, what can you buy after a year of saving $100 in the bank that 

will be? 1. more than a year ago, 2. as much as a year ago, 3. less than a year ago, 4. impossible to 

calculate". "Which do you think is riskier in general, equity or debt funds? 1. equity funds, 2. debt funds, 

3. never heard of equity funds, 4. never heard of debt funds, 5. neither, 6. the same". Which do you think 

is riskier in general, Main Board stocks or GEM stocks?1. Main Board, 2. GEM, 3. never heard of Main 

Board stocks, 4. never heard of GEM stocks, 5. Neither heard of them, 6. the same ". If the answer to the 

above question is " impossible to calculate " or " never heard of either", the value is 0. ]. Those questions cover the financial calculating ability, the knowledge about inflation and risk, and financial market knowledge of the sample. The validity and reliability of the resulting scale were then tested based on the responses to these questions. The Cronbach's alpha of these scales was 0.865, the KMO value was 0.786, and the Bartlett's test was significant; Besides, the component matrix loadings of each rotation were greater than 0.682, thus the validity and reliability of above scales measuring financial literacy. While, we quantify market expectations (expectmacroiC) with the expectation of residents to the employment market, real estate market and economic growth, which is obtained from questionnaire items. 

8.There is no discussion of diagnostic statistics used in data analysis section. Author must incorporate various diagnostic statistics methods in order to understand various characteristics of data used for analysis. 

Response:

Thank you for your comment. We supplement the diagnostic statistics used in data analysis section as follows.

Table 1 shows the characteristics of samples. 50.90% of the questionnaire respondents are female. The respondents have an average age of 49.07, of which range from 46 to 60 years old, accounting 24.89% . Nearly 43.71% of the respondents have 9 years of schooling or below, 46.73% of the respondents have attended a high school (10-12 years), and less than 10% of people have 12 years of schooling and beyond. About 10.38% of interviewees stated that their household annual income is less than 30,000 Yuan, and 47.68% of them have an annual income above 90,000 Yuan. This Demographic characteristic respondents is basicly consitent with the data of China, indicating our data has representativity.

Table1 Demographic profiles of the respondents

Item Category Proportion

Gender Male 49.10%

 Female 50.90%

Age(years) <25 23.45%

 25-35 12.36%

 36-45 12.06%

 46-60 24.89%

 >60 27.24%

Schooling years 0-9years 43.71%

 10-12years 46.73%

 >12years 9.56%

Household annual income

(thousand yuan) <30 10.38%

 30-60 10.26%

 60-90 31.67%

 >90 47.68%

9.Discussion in the data analysis section should be supported with references from literature review and acceptance and rejection of hypothesis should be properly highlighted. 

Response:

Thank you for your professional comment. Based on your suggestion, we have modified our version by adding more literature review to support our empirical conclusion. Meanwhile, we clarify the proof of all empirical analyses.

10.There should be uniformity in the reference style given in references section of the research study. 

Response:

Thanks for your rigorous attitude to writing. We have harmonized the reference style given in references section

Overall, we thank you for helping us improve this manuscript. We appreciate your advice and the time you have devoted to our study. Best wishes!

---

## [Decision Letter · Decision Letter 1]

1 Feb 2024

The peer effects of resident stock market participation: Evidence from 2019 CHFS Household Finance Survey

PONE-D-23-25473R1

Dear Dr. Kong,

We’re pleased to inform you that your manuscript has been judged scientifically suitable for publication and will be formally accepted for publication once it meets all outstanding technical requirements.

Kind regards,

Wajid Khan

Academic Editor

PLOS ONE

Additional Editor Comments (optional):

Reviewers' comments:

Reviewer's Responses to Questions

**Comments to the Author**

1. If the authors have adequately addressed your comments raised in a previous round of review and you feel that this manuscript is now acceptable for publication, you may indicate that here to bypass the “Comments to the Author” section, enter your conflict of interest statement in the “Confidential to Editor” section, and submit your "Accept" recommendation.

Reviewer #1: All comments have been addressed

Reviewer #2: All comments have been addressed

2. Is the manuscript technically sound, and do the data support the conclusions?

Reviewer #1: Yes

Reviewer #2: Yes

3. Has the statistical analysis been performed appropriately and rigorously? 

Reviewer #1: Yes

Reviewer #2: Yes

4. Have the authors made all data underlying the findings in their manuscript fully available?

Reviewer #1: No

Reviewer #2: Yes

5. Is the manuscript presented in an intelligible fashion and written in standard English?

Reviewer #1: Yes

Reviewer #2: Yes

6. Review Comments to the Author

Reviewer #1: I have no further comments. I am satisfied with the revised paper. I do have three minor suggestions:

1. Consider giving the paper a more appealing title.

2. It should be acceptable for CHFS to share the data for research purposes.

3. Put in some effort to improve the language used in the paper.

Reviewer #2: All changes suggested by reviewers have been incorporated.

The editor should further proceed with publication pf paper

7. PLOS authors have the option to publish the peer review history of their article (what does this mean?). If published, this will include your full peer review and any attached files.

Reviewer #1: **Yes: **Cheng Yuan

Reviewer #2: No

---

## [Editor Report · Acceptance letter]

1 Apr 2024

PONE-D-23-25473R1 

PLOS ONE

Dear Dr. Kong, 

I'm pleased to inform you that your manuscript has been deemed suitable for publication in PLOS ONE. Congratulations! Your manuscript is now being handed over to our production team.

Kind regards, 

on behalf of

Dr. Wajid Khan 

Academic Editor

PLOS ONE